# Study of TiO_2_ on the Voltage Holdoff Capacity of Cr/Mn-Doped Al_2_O_3_ Ceramic in Vacuum

**DOI:** 10.3390/ma16145048

**Published:** 2023-07-17

**Authors:** Dandan Feng, Xiaojing Wang, Shike Zhao, Baipeng Song, Guanjun Zhang

**Affiliations:** 1Beijing Vacuum Electronics Research Institute, Beijing 100015, China; xiaojingwang129@163.com (X.W.); shikezhao1966@163.com (S.Z.); 2State Key Laboratory of Electrical Insulation and Power Equipment, Xi’an Jiaotong University, Xi’an 710049, China; bpsong@xjtu.edu.cn (B.S.); gjzhang67@126.com (G.Z.)

**Keywords:** Al_2_O_3_ ceramic, TiO_2_, secondary electron yield, surface resistivity, surface flashover threshold

## Abstract

With the development of vacuum electronic devices toward high power, high frequency, and miniaturization, the voltage holdoff capacity of the insulation materials in devices has also been raised to a higher demand. Cr/Mn/Ti-doped Al_2_O_3_ ceramics were prepared, and the bulk density, micromorphology, phase composition, resistivity, secondary electron emission coefficient, and surface flashover threshold in the vacuum of the Al_2_O_3_ were characterized. The results show that the addition of TiO_2_ to the Al_2_O_3_ ceramic can promote the sintering of the ceramic. The Cr/Mn/Ti-doped Al_2_O_3_ ceramic with a homogeneous microstructure can be obtained by an appropriate amount of TiO_2_ addition. In the process of the heat treatment, the TiO_2_ in the ceramics was reduced to a certain degree, which had an impact on the microstructure of the Al_2_O_3_ ceramic. Adding a small amount of TiO_2_ can improve the voltage holdoff performance in the vacuum. The value of the surface flashover threshold in the vacuum of the Cr/Mn/Ti-doped Al_2_O_3_ ceramic containing 1 wt.% TiO_2_ reached a value of 33 kV, which is 32% higher than that of the basic Al_2_O_3_ ceramic. The preparation of Al_2_O_3_ ceramics with a high voltage holdoff capacity in a vacuum provides fundamental technical support for the development of vacuum electronic devices.

## 1. Introduction

Al_2_O_3_ ceramics play a significant role in vacuum electronic devices, such as high-voltage insulation, vacuum sealing, power transmission, and support fixation [1,2]. Studies have shown that as a solid insulator, the voltage holdoff capacity of Al_2_O_3_ ceramics in a vacuum is often lower than vacuum gaps of the same size. The surface of the insulator is the weakest point in vacuum-insulation systems because its bulk voltage holdoff capacity is generally greater than the vacuum gap of the same size [3,4]. Surface flashover occurs in many vacuum electronic devices when the applied voltage exceeds a certain value, resulting in the failure of or damage to the device [5,6,7,8,9,10]. With the development of vacuum electronic devices toward high power, high frequency, and miniaturization, such as high-power klystron and high-power pseudo spark switch, the operating voltage of the device increases exponentially, making surface flashover of Al_2_O_3_ ceramics become an important factor affecting their reliability and restricting the development of vacuum electronic devices [11,12,13,14,15]. Therefore, improving the surface voltage holdoff capacity of Al_2_O_3_ ceramics has become one of the urgent problems to be solved in the field of vacuum electronics.

Surface flashover is related to many factors, such as the materials’ characteristics, shape structure, and surface roughness of the ceramic. According to secondary electron emission avalanche theory (SEEA) [16,17,18], the initiation of a surface flashover is usually started by the emission of electrons (generally by field emission or thermal field emission) from the cathode triple junction (CTJ—the interface where the insulator, cathode, and vacuum are in close proximity). Some of the electrons impact the surface of the insulator, producing additional electrons by secondary emission. Some of these secondary electrons will again strike the surface, producing tertiary electrons. Continuation of this process results in the development of an SEEA. The final stages of surface flashover are predominantly thought to occur in desorbed surface gas or in vaporized insulator material. Hence, the secondary electron yield (SEY) of the ceramic is an important factor affecting its surface flashover voltage. Therefore, the SEY of the ceramic should be reduced as much as possible to improve its surface flashover voltage. In addition, the high surface resistivity of the ceramic is averse to charge leakage, which will increase the strength of the local electric field, leading to the occurrence of surface flashover.

Researchers have improved the surface voltage holdoff capacity of Al_2_O_3_ ceramics through surface modification and the bulk doping method. Tangal S. Sudarshan et al. [19] reduced the SEY of Al_2_O_3_ ceramics and improved their surface flashover voltage in a vacuum by coating Cr_2_O_3_ on the surface. Yamamoto O. et al. [20] studied the influence of surface roughness on the surface voltage holdoff capacity of dielectric materials and improved the surface flashover voltage of Al_2_O_3_ ceramics by roughening the surface. Zheng Jiagui et al. [21] reduced the SEY of Al_2_O_3_ ceramics to 2~3 by coating Cr, Mn, and Ti on the surface. Zhang Hao et al. [22] improved the surface flashover voltage of a glass ceramic by adding Cr_2_O_3_ to it. However, the mechanism of surface flashover of insulating materials in a vacuum is complex, and the research content involves multidisciplines, which are closely related to the composition and microstructure of materials. Therefore, further systematic and in-depth research on ceramic materials with a high-voltage holdoff capacity is needed to provide key basic technology support for the development of vacuum electronic devices toward high power, high frequency, and miniaturization.

Among the methods to improve the voltage holdoff capacity of insulating materials in a vacuum, surface modification can improve the surface properties without changing the intrinsic structure of the ceramic, thus increasing its surface flashover voltage. However, this method has large structural limitations, and the evaporation of coatings within the device is difficult to control. Bulk doping is easier to achieve for some special structures, and the material performance is relatively stable. The SEY of Cr_2_O_3_ [16] is very low, and MnO_2_ has low resistivity. They are usually used to improve the voltage holdoff capacity of Al_2_O_3_ ceramics as additives. The properties of Cr_2_O_3_ and MnO_2_-doped Al_2_O_3_ ceramics were studied previously by our research group. The results show that the SEY of Al_2_O_3_ ceramics can be reduced effectively by adding Cr_2_O_3_ to the ceramics, while the surface resistivity of Al_2_O_3_ ceramics can be reduced by one order of magnitude with the addition of MnO_2_. However, with the increase in the addition of MnO_2_, the grain size of the Al_2_O_3_ ceramic increases, the bulk density of the Al_2_O_3_ ceramic decreases, and the uniformity of the microstructure becomes worse. In this paper, in order to further reduce the resistivity of the Al_2_O_3_ ceramic, on the basis of the previous research on Cr_2_O_3_ and MnO_2_-doped Al_2_O_3_ ceramics, we chose TiO_2_, with semiconductor properties, as an additive, and studied the effect of the TiO_2_ addition on the performance of the Al_2_O_3_ ceramics. TiO_2_ has low resistivity, but few studies have been reported on the effects of TiO_2_ on the resistivity and voltage holdoff capacity of Al_2_O_3_ ceramics, since the valence of Ti changes in different compounds and its mechanism of action in materials is complicated [23,24]. On the basis of Cr/Mn-doped Al_2_O_3_ ceramics, we fixed Cr_2_O_3_:MnO_2_ = 1:1 (mass ratio) in this paper and studied the effects of the additional amount of TiO_2_ on the microstructure, phase composition, secondary electron emission characteristics, resistivity, and surface flash properties of the Al_2_O_3_ ceramic in a vacuum.

## 2. Experimental Procedures

### 2.1. Samples Preparation

The 95% Al_2_O_3_ ceramics were chosen as fundamental Al_2_O_3_ ceramic materials. SiO_2_ and CaO were introduced in the form of silica and calcium carbonate as additives to promote the liquid phase sintering of Al_2_O_3_ ceramics. Cr_2_O_3_, MnO_2,_ and TiO_2_ were added to the ceramic as additives. The Al_2_O_3_ ceramics were prepared from high-purity raw materials (≥99.9%). The ceramic powders were prepared by spray granulation method, then were formed by cold isostatic pressing. After that, the green ceramic bodies were sintered under air atmosphere using a high-temperature muffle furnace. Figure 1 shows the preparation process of the Cr/Mn/Ti-doped Al_2_O_3_ ceramics. The parameters of the sintering process are shown in Table 1.

The prepared ceramics were processed into samples with dimensions of Φ 26 mm × 5 mm, Φ 26 mm × 1 mm, and Φ 40 mm × 2 mm via grinding for testing the surface flashover voltage in a vacuum, secondary electron emission characteristics, and resistivity, respectively. The samples were numbered according to the amount of TiO_2_ added, as shown in Table 2, and photos of the samples are shown in Figure 2.

A portion of samples were loaded into a high-temperature hydrogen furnace and heat treatment was performed at 1450 °C under wet hydrogen conditions to study the effect of the heat treatment process on the properties of the ceramics, considering the requirements of metallization and welding when applied in vacuum electronic devices. The parameters of the heat treatment process are shown in Table 3.

### 2.2. Samples Characterization

#### 2.2.1. Bulk Density, Resistivity, and Microstructure Characterization

The bulk density of the samples was measured by Archimedes’ drainage method. High resistance meter (SM7120, HITACHI, Tokyo, Japan) was used to measure the surface resistivity and volume resistivity of the samples. Scanning electron microscope (SU3800, HITACHI, Tokyo, Japan) was applied to observe the microstructure of the samples. The microscopic composition of the samples was analyzed using X-ray diffractometer (D8 Advance, BRUKER, Karlsruhe, Germany) and X-ray photoelectron spectrometer (SM7120, HITACHI, Tokyo, Japan).

#### 2.2.2. Test of Secondary Electron Emission Characteristics

There will be charges accumulated on the surface when a ceramic is bombarded by electrons. In order to eliminate the influence of surface charge accumulation, the three-gun method was applied to measure the SEY of the ceramic materials. A diagrammatic sketch of the test system’s structure is shown in Figure 3. The system works as follows: the first charge neutralizer gun, which is a low-energy electron gun, is activated to remove the positive surface charges when they appear on the ceramic surface. The second charge neutralizer gun is activated to remove the negative surface charges when they appear on the ceramic surface. The surface potential of the ceramics is stabilized to ground potential by the cooperation of two neutralizer guns to eliminate the charge accumulated on the ceramic surface, in order to ensure the accuracy of secondary electron emission testing [25]. During the secondary electron emission test, the incident electron beam is perpendicular to the sample; the primary electron energy range is 0~3500 eV; the incident current is 0.1 μA with a pulse width of 5 μs; the temperature during the test is kept as room temperature; and the vacuum degree of the cavity is lower than 1 × 10^−5^ Pa.

#### 2.2.3. Test of Surface Flashover Threshold in Vacuum

A DC high-voltage vacuum test system was used to test the surface flashover threshold of Al_2_O_3_ ceramics in a vacuum, and the diagrammatic sketch of the testing device is shown in Figure 4a. The electrodes were disc-shaped flat shape made of stainless steel, with a diameter of 66 mm, and a distance between electrodes of the sample height, i.e., 5 mm. A photo of the flat electrode and test structure is shown in Figure 4b. The test started when the vacuum degree of the chamber reached 1 × 10^−4^ Pa. During the test, a negative polarity DC voltage with the linear rise rate of 500 V/s was applied to the sample until the surface flashover occurred. Then, we gradually reduced the voltage until the surface flashover completely disappeared. If this voltage is repeatedly applied to the sample 3 times, and the sample can avoid flashover and remain stable, then the voltage was recorded as the surface flashover threshold in a vacuum. The voltage holdoff capacity of Al_2_O_3_ ceramics in a vacuum is represented by the surface flashover threshold, denoted by U_ho_. Five samples were tested per group, and the average value of U_ho_ was calculated as the final result.

## 3. Results and Discussions

### 3.1. Bulk Density and Micromorphology

The influence of TiO_2_ on the bulk density and micromorphology of the Al_2_O_3_ ceramics was studied. It can be seen from Table 4 that we can increase the bulk density of the ceramics by adding a little amount of TiO_2_ to Cr/Mn-doped Al_2_O_3_ ceramics. The bulk density of the Al_2_O_3_ ceramics with 1 wt.% TiO_2_ is highest, reaching 3.810 g/cm^3^. After that, the bulk density of the Al_2_O_3_ ceramics decreases with the increase in the content of TiO_2_. The addition of TiO_2_ to the Al_2_O_3_ ceramics has an effect on their sintering properties. TiO_2_ can form the finite solid solution with Al_2_O_3_ when it is added to ceramics. In the range of 1300 °C–1700 °C, the solid solubility of TiO_2_ in the α- Al_2_O_3_ ceramics is 0.27 wt.% [26]. The TiO_2_ residual is located at the grain boundary, in which case it can react with MnO_2_ and form a low-temperature eutectic mixture. The low eutectic presents a liquid phase during sintering, which reduces the sintering temperature of Al_2_O_3_ ceramics [27]. Secondly, in a TiO_2_-Al_2_O_3_ solid solution, Ti^4+^ replaces Al^3+^, which produces lattice deformation and a large number of cation vacancies. The energy potential barrier required for lattice particle diffusion is reduced and the lattice is activated, which is conducive to the diffusion and transfer of substances and promotes the sintering of ceramics.

Figure 5 shows the fractured surface of the Cr/Mn/Ti-doped Al_2_O_3_ ceramics. We can find from Figure 5 that there is little difference in the microstructures between CMT05 and CMT10, and there are mainly elongated grains, with an average particle size of about 8–10 μm. As the amount of TiO_2_ continued to increase, the grain size of the CMT20 and CMT30 samples increased, and large grains appeared locally, as shown in Figure 5c,d. The addition of excessive TiO_2_ to ceramics makes a local abnormal grain growth during the ceramic sintering process. The pores in the ceramic cannot be discharged promptly, which is likely to cause pores wrapped up (as circled in the Figure 5) in the interior and boundary of the grains, resulting in the reduction in the bulk density of the ceramics, and the uniformity of the microstructure will also become worse.

### 3.2. Microstructure and Phase Composition

XPS and XRD were applied to analyze the microstructure and phase composition of the Al_2_O_3_ ceramics. Since Ti is a variable valence element, its elemental valence and phase composition in the ceramics may change during the sintering and heat treatment. Figure 6 shows the XPS pattern of CMT30. The XPS pattern of Ti 2p in Figure 6b shows that the binding energy of the Ti 2p_1/2_ energy level of Ti^4+^ in the untreated samples is 464.4 eV, and the binding energy of the Ti 2p_3/2_ energy level is 459.2 eV [28,29]. After the heat treatment, the binding energy of the Ti 2penergy level of the samples moves toward the direction of lower energy, indicating that hydrogen has a certain degree of reduction on TiO_2_ during the heat treatment.

Figure 7 and Figure 8 show the XRD patterns of the Al_2_O_3_ ceramics. It can be seen that the basic Al_2_O_3_ ceramics A-0 only contain the phase of Al_2_O_3_. The locally enlarged image in Figure 7b shows that the characteristic peak of the Al_2_O_3_ at 2 Theta = 35.15 shifts toward a small angle direction. This is due to the solid solution of TiO_2_ and Al_2_O_3_. Ti^4+^ replaces Al^3+^, which changes the cell parameters of Al_2_O_3_. Combined with Figure 8, we can find that in addition to the main crystal phase of Al_2_O_3_, the CMT30 sample also contains a small amount of other crystal phases such as CaAl_2_Si_2_O_8_, Al_2_SiO_5,_ and MnTiO_3_, which are generated by the reaction of Al_2_O, CaO, SiO_2_, MnO_2,_ and TiO_2_ in the ceramics. Figure 8 is the particle comparison of the XRD data of the A-0 and CMT30 samples after normalization, displaying that the CMT30 sample before and after the heat treatment contained both TiO_2_ and TiO_0.5_. And the content of the TiO_2_ phase decreased and the content of the TiO_0.5_ phase increased in CMT30 after the heat treatment. In addition, after the heat treatment, the Ti_8_O_15_ phase appeared in CMT30. Both the XPS and XRD results show that TiO_2_ in Al_2_O_3_ ceramics was reduced during the heat treatment process, resulting in an increase in the content of the anoxic phase in the Cr/Mn/Ti-doped Al_2_O_3_ ceramics.

### 3.3. Secondary Electron Emission Characteristics

According to the SEEA theory [16,17,18], secondary electron emission of ceramics is a factor which influences the surface flashover voltage directly. We measured the SEY of the Al_2_O_3_ ceramics with different contents of TiO_2_. The secondary electron emission curves and maximum value of the SEY are shown in Figure 9 and Table 5, respectively.

The results show that the SEY of the Cr/Mn- or Cr/Mn/Ti-doped Al_2_O_3_ ceramics is significantly lower than that of the basic Al_2_O_3_ ceramics, because Cr_2_O_3_ with a low secondary electron emission coefficient as an additive can effectively reduce the SEY of the Al_2_O_3_ ceramics. While the SEY of the ceramics increases with the increase in the content of TiO_2_. Particularly, the SEY of the ceramics increases significantly when the content of TiO_2_ is more than 2 wt.%. This is mainly attributed to the effect of TiO_2_ on the microstructure of the Al_2_O_3_ ceramics. Adding TiO_2_ to the Al_2_O_3_ ceramics can promote the process of sintering. The grain size of the ceramics increases significantly when the content of TiO_2_ is more than 2 wt.%.

According to the SEEA theory, the primary electrons are accelerated by the electric field and gain energy, then they collide with the surface of the ceramic and produce the secondary electrons. The secondary electrons collide with the surface again under the effect of the electric field, after they escape from the surface. The whole process is repeated and the number of electrons multiplies rapidly, eventually leading to the occurrence of the electron avalanche. Electron avalanches cause the release of adsorbed gas on the ceramic surface, which is ionized by high-energy electrons, producing a large amount of plasma. Eventually, surface flashover occurs. Usually, the main factors affecting the collisional ionization process are the energy of the electron before the collision and the ionization energy of the collided lattice molecule or atom. The collisional ionization coefficient α is commonly used to describe the collisional ionization. The collisional ionization coefficient of an electron is the number of collisional ionizations produced by an electron traveling a unit distance under the action of the electric field. And the collisional ionization coefficient can generally be expressed as follows:(1)$$\alpha =Aexp(-\frac{BV_{i}}{\lambda E})$$
where *V_i_* is the ionization energy of the collided lattice molecule or atom, *λ* is the mean free path of the electron, and *E* is the electric field strength. *A*, *B* are constants related to materials. It can be found from the expression of the collisional ionization coefficient that the mean free path of the electron influences the coefficient directly. A longer mean free path of the electron means that the electron is subjected to the electric field force between the two collisions for a longer time under the effect of the electric field, making ionization easy to occur. The grain size of the ceramics increases significantly when the content of TiO_2_ is more than 2 wt.%. And the mean free path of the secondary electrons increases in the process of escape. The ionization coefficient of recollision ionization also becomes larger, which makes ionization easier to occur, resulting in the increase in the SEY.

### 3.4. Surface and Volume Resistivity 

The surface resistivity of the ceramics is another important factor that affects the voltage holdoff capacity in a vacuum. The high surface resistivity of the ceramics is not conducive to charge leakage, resulting in an increase in the local field strength. And the surface flashover is aggravated. Thus, the surface resistivity of the ceramics must be reduced to a certain extent in order to improve the voltage holdoff capacity. It can be seen from the resistivity of the Al_2_O_3_ ceramics in Table 6 that the resistivity of the Cr/Mn-doped Al_2_O_3_ ceramics did not change a lot before and after the heat treatment, while the resistivity of the Al_2_O_3_ ceramics with TiO_2_ changed significantly.

When TiO_2_ was added to the Cr/Mn-doped Al_2_O_3_ ceramics, the change in the resistivity of the ceramics was not significant before the heat treatment, and the surface resistance of the ceramics was one order of magnitude lower than that of the basic Al_2_O_3_ ceramics. With the increase in the content of TiO_2_, the surface resistivity of the ceramics did not change much. The volume resistivity of the CMT20 and CMT30 was reduced by two orders of magnitude compared with the base Al_2_O_3_ ceramics A-0. After the heat treatment, the resistivity of the Cr/Mn/Ti-doped Al_2_O_3_ ceramics decreased significantly. The surface resistivity and volume resistivity of the CMT10 decreased by 4 and 5 orders of magnitude compared with the base Al_2_O_3_ ceramics, reaching 7.6 × 10^11^ Ω and 3.47 × 10^11^ Ω·cm, respectively. The resistivity exceeded the measuring range of the high resistance meter (˂10^8^ Ω) when the content of TiO_2_ was more than 3 wt.%. The color of the samples changed significantly after the heat treatment, as shown in Figure 10.

According to the XPS results in Figure 6, the binding energy of the Ti 2p level in the sample moved to the direction of low energy after the heat treatment. TiO_2_ was reduced by hydrogen to a certain extent, and the O:Ti ratio declined. The XRD analysis results show, meanwhile, that the content of the anoxic phase in the ceramics increased after the heat treatment. According to the relevant literature [30,31], the formation of anoxic phase Ti_n_O_2n−1_ will significantly reduce the resistivity of the ceramics. And the higher the content of the anoxic phase, the darker the ceramic color and the smaller the resistivity are. This is also consistent with the pattern of the samples’ color in the experiment.

### 3.5. Voltage Holdoff Capacity in Vacuum

We measured the surface flashover threshold of the Al_2_O_3_ ceramics, and the size of the samples was Φ 26 mm × 5 mm. The surface flashover threshold of the samples before and after the heat treatment were tested by experiments, and the results are shown in Table 7. 

As shown in Table 7, the surface flashover threshold slightly increased when a small amount of TiO_2_ was added to the ceramics. However, the surface flashover threshold of the ceramics changed significantly after the heat treatment. The effect of the heat treatment process on the Al_2_O_3_ ceramics with the TiO_2_ additive was different from the basic Al_2_O_3_ ceramics A-0 and the Cr/Mn-doped Al_2_O_3_ ceramics CM11. The U_ho_ of the Cr/Mn/Ti-doped Al_2_O_3_ ceramics with a small amount of TiO_2_ was significantly improved after the heat treatment. The U_ho_ of CMT11 reached 33 kV, which was 32% higher than that of the basic Al_2_O_3_ ceramics. After the heat treatment with wet hydrogen, the U_ho_ of the basic Al_2_O_3_ ceramics slightly decreased, while the U_ho_ of the Cr/Mn/Ti-doped Al_2_O_3_ ceramics increased significantly. The U_ho_ of the Cr/Mn/Ti-doped Al_2_O_3_ ceramics in a vacuum is higher than that of the base Al_2_O_3_ ceramics, which is mainly attributed to the following three aspects. Firstly, the SEY of the Cr/Mn/Ti-doped Al_2_O_3_ ceramics is obviously lower than that of the basic Al_2_O_3_ ceramics, which makes the secondary electron avalanche hard to happen on the surface of the ceramics when the voltage is applied; thus, the surface flashover threshold is larger. Secondly, the surface resistivity of the sample with TiO_2_ did not change a lot compared with the basic Al_2_O_3_ ceramics before the heat treatment, but the surface resistivity and volume resistivity of the Cr/Mn/Ti-doped Al_2_O_3_ ceramics were significantly lower than those of the basic Al_2_O_3_ ceramics after the heat treatment. The low surface resistivity of the ceramics can make the surface charge be conducted timely and effectively, so the probability of the flashover occurrence along the surface can be reduced. Finally, the bulk density and microstructure uniformity of the ceramics can be improved with a small amount of TiO_2_ addition, which can increase the stability of the ceramics in the process of the applied voltage.

According to the data of the surface flashover threshold in a vacuum from Table 7, when the content of TiO_2_ was more than 1 wt.%, the U_ho_ of CMT20 and CMT30 were very low after the heat treatment, which were 18 kV and 15 kV, respectively. This is because the resistivity of the ceramics after the heat treatment with wet hydrogen was too low. When the voltage was applied to the ceramics, there was a large leakage current, resulting in the action of power failure protection, and the power of the high-voltage equipment was cut. So, the test value of the surface flashover threshold was very low.

## 4. Conclusions

In this paper, Al_2_O_3_ ceramics with a high surface flashover threshold in a vacuum were prepared by bulk doping Cr_2_O_3_, MnO_2,_ and TiO_2_, and the effect of the content of TiO_2_ on the properties of the Al_2_O_3_ ceramics was studied. The main conclusions are as follows:(1)The addition of TiO_2_ to the Al_2_O_3_ ceramics can promote the sintering of the ceramics, and ceramics with a high volume density and uniform microstructure can be prepared by an appropriate additive amount of TiO_2_.(2)TiO_2_ in the ceramics was reduced during the heat treatment with wet hydrogen condition. The resistivity of the Cr/Mn/Ti-doped Al_2_O_3_ ceramics decreased significantly with the appearance of the anoxic phase.(3)The microstructure of the Cr/Mn/Ti-doped Al_2_O_3_ ceramics with 1 wt.% TiO_2_ is uniform and compact, and the surface resistivity of the sample is reduced to 7.6 × 10^11^ Ω. The surface flashover threshold in a vacuum reaches 33 kV, which is 32% higher than that of the basic Al_2_O_3_ ceramics.(4)The resistivity of the ceramics will be too low when the content of TiO_2_ exceeds 2 wt.%, resulting in a large leakage flow and low surface flashover threshold during the experiment.

## Figures and Tables

**Figure 1 materials-16-05048-f001:**
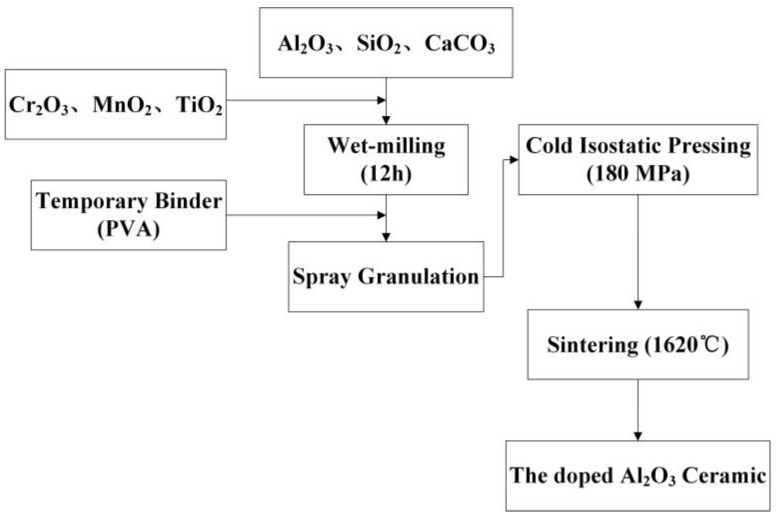
Preparation process of the Al_2_O_3_ ceramics samples.

**Figure 2 materials-16-05048-f002:**
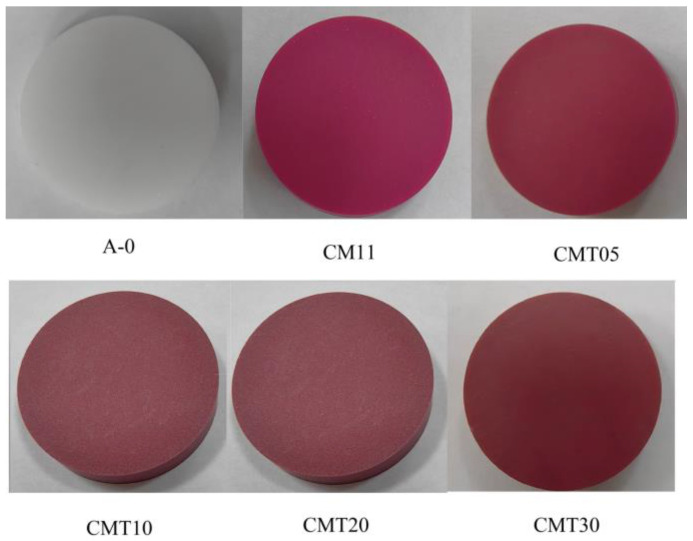
Photos of the Al_2_O_3_ ceramics before heat treatment.

**Figure 3 materials-16-05048-f003:**
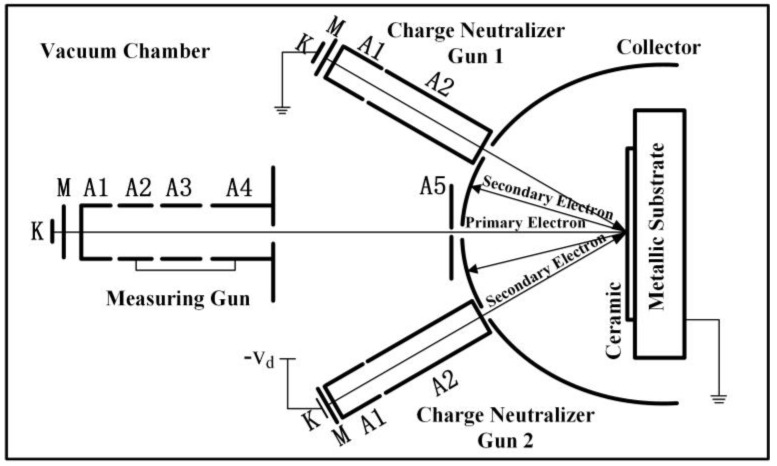
Diagrammatic sketch of secondary electrons emission characteristics test system.

**Figure 4 materials-16-05048-f004:**
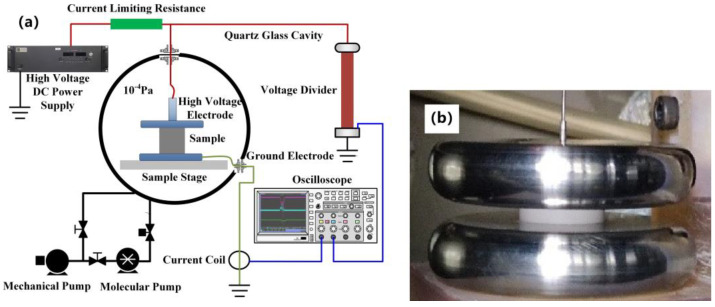
(**a**) Diagrammatic sketch of the testing device of surface flashover threshold in vacuum; (**b**) Photo of the flat electrode and test structure.

**Figure 5 materials-16-05048-f005:**
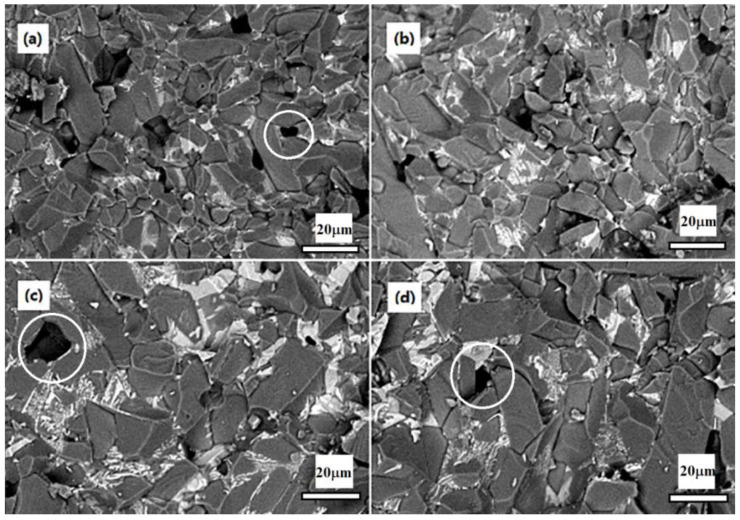
SEM of the doped Al_2_O_3_ ceramics: (**a**) CMT05; (**b**) CMT10; (**c**) CMT20; (**d**) CMT30.

**Figure 6 materials-16-05048-f006:**
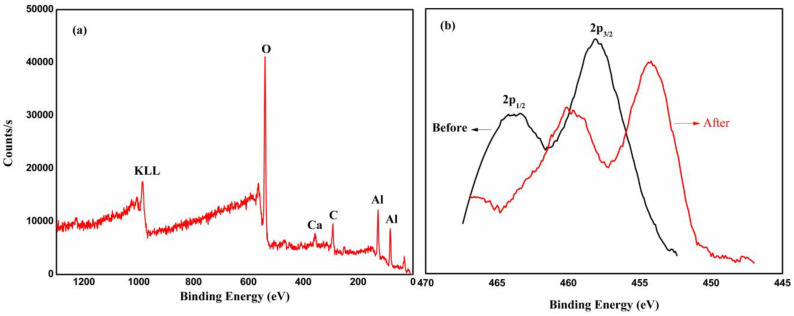
XPS pattern of CMT30: (**a**) full spectrum; (**b**) XPS pattern of Ti 2p of the sample before/after heat treatment.

**Figure 7 materials-16-05048-f007:**
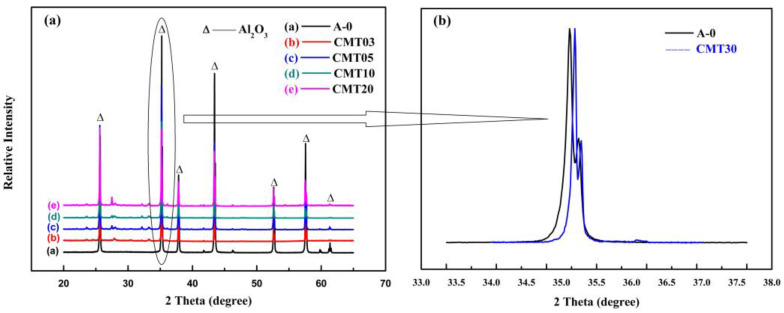
XRD pattern of the Al_2_O_3_ ceramics with different content of TiO_2_: (**a**) Full spectrum; (**b**) Partial enlarge view of A-0 and CMT30.

**Figure 8 materials-16-05048-f008:**
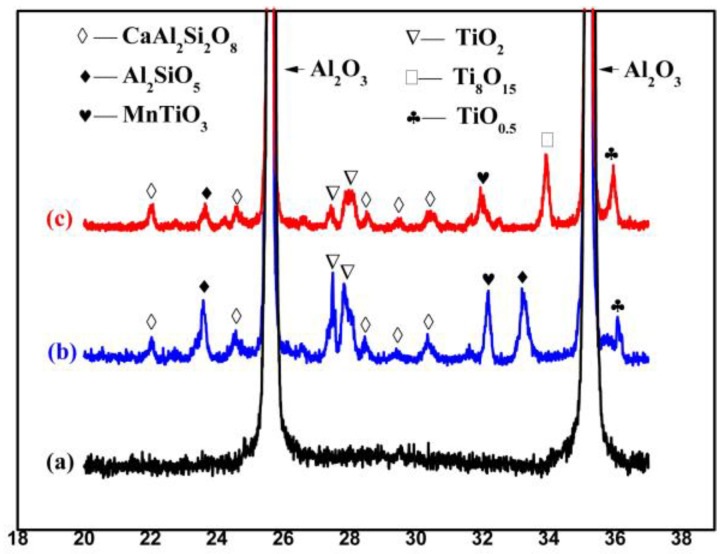
Partial comparison of XRD pattern: (**a**) A-0; (**b**) CMT30 before heat treatment; (**c**) CMT30 after heat treatment.

**Figure 9 materials-16-05048-f009:**
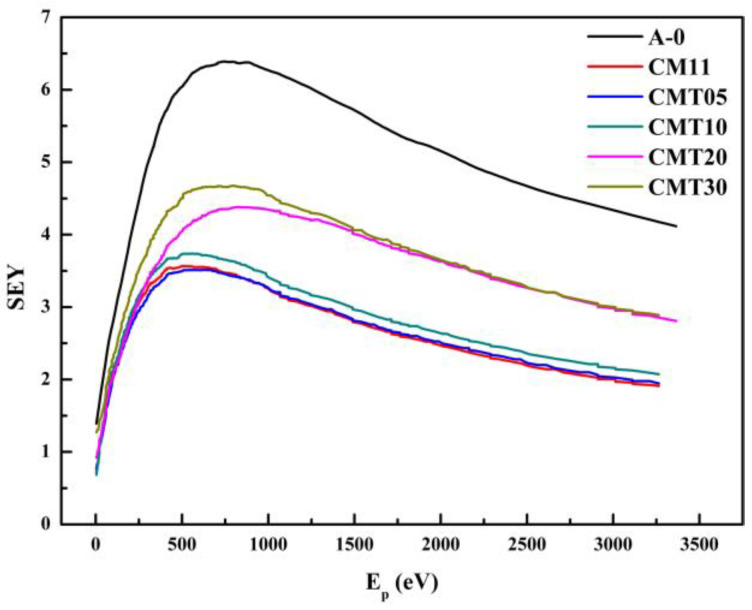
Secondary electron yield (SEY) curves of the Al_2_O_3_ ceramics with different content of TiO_2_.

**Figure 10 materials-16-05048-f010:**
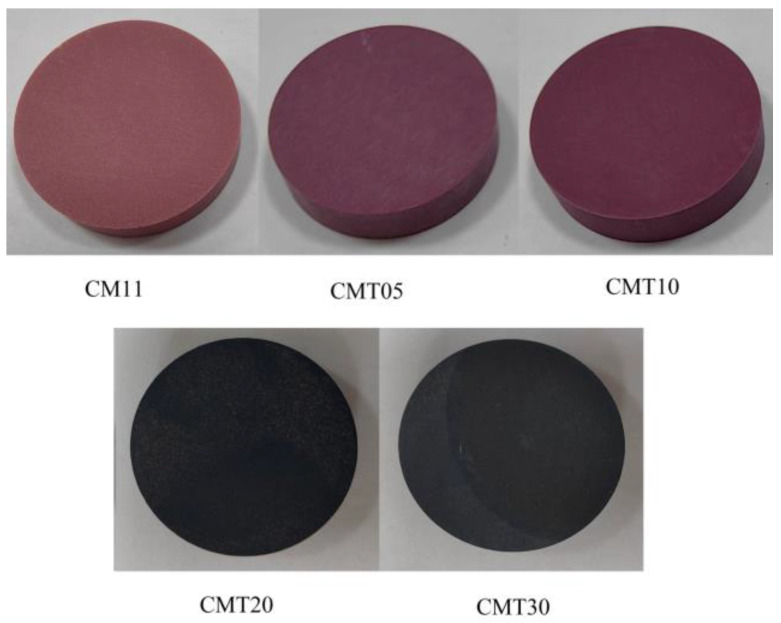
Photos of the Cr/Mn/Ti-doped Al_2_O_3_ ceramics after heat treatment.

**Table 1 materials-16-05048-t001:** Sintering parameters of the Al_2_O_3_ ceramics samples.

Temperature, °C	25–500	500–500	500–1300	1300–1620	1620–1620
Time, min	297	120	267	200	240

**Table 2 materials-16-05048-t002:** Sample number and addition amount of TiO_2_.

Sample Number	Cr_2_O_3_:MnO_2_	TiO_2_, wt.%
A-0	-	-
CM11	1:1	-
CMT05	1:1	0.5
CMT10	1:1	1
CMT20	1:1	2
CMT30	1:1	3

**Table 3 materials-16-05048-t003:** Parameters of heat treatment process.

Temperature, °C	25–1000	1000–1000	1000–1450	1450–1450	1450–1000	1000–200	200–25
Time, min	85	20	100	60	30	furnace cooling	
Atmosphere	Wet hydrogen, dew point 26–27 °C						Dry nitrogen

**Table 4 materials-16-05048-t004:** Bulk density of Al_2_O_3_ ceramics with different content of TiO_2_.

Sample Number	Bulk Density, g/cm^3^
A-0	3.75
CM11	3.782
CMT05	3.782
CMT10	3.810
CMT20	3.744
CMT30	3.716

**Table 5 materials-16-05048-t005:** Maximum value of SEY of the Al_2_O_3_ ceramics with different content of TiO_2_.

Sample Number	SEEY
A-0	6.388
CM11	3.566
CMT05	3.513
CMT10	3.739
CMT20	4.379
CMT30	4.675

**Table 6 materials-16-05048-t006:** Resistivity of the Al_2_O_3_ ceramics with different content of TiO_2_.

Sample Number	Surface Resistivity, Ω		Volume Resistivity, Ω·cm	
	Before Heat Treatment	After Heat Treatment	Before Heat Treatment	After Heat Treatment
A-0	7.405 × 10^15^	4.854 × 10^15^	2.35 × 10^16^	4.353 × 10^16^
CM11	7.21 × 10^14^	6.72 × 10^14^	3.81 × 10^15^	4.65 × 10^15^
CMT05	7.23 × 10^14^	5.236 × 10^14^	7.875 × 10^15^	8.254 × 10^14^
CMT10	4.875 × 10^14^	7.6 × 10^11^	4.10 × 10^15^	3.47 × 10^11^
CMT20	1.77 × 10^14^	6.68 × 10^8^	2.06 × 10^14^	1.41 × 10^10^
CMT30	1.57 × 10^14^	˂10^8^	2.96 × 10^14^	˂10^8^

**Table 7 materials-16-05048-t007:** Value of surface flashover threshold in vacuum of the Al_2_O_3_ ceramics.

Sample Number	U_ho_, kV	
	Before Heat Treatment	After Heat Treatment
A-0	27	25
CM11	29	28
CMT05	30	28
CMT10	30	33
CMT20	28	18
CMT30	27	15

## Data Availability

Data available on request due to restrictions eg privacy or ethical.

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
