# Peer review of "Study of TiO2 on the Voltage Holdoff Capacity of Cr/Mn-Doped Al2O3 Ceramic in Vacuum"

_materials, 2023, doi:10.3390/ma16145048_

Round 1
Reviewer 1 Report
The proposed topic interests the scientific materials community, the methodology is adequate for the research purpose, and the results are appropriately discussed. However, some inaccuracies need to solve
It is stated that SEY is shown in Table 5 and Figure 9, but the data presented in both are different, especially in CMT03 and CTM30 samples.
The Figure 4 label refers to (a) and (b) labels, which are not in the figure.
Please clarify how many samples were studied and state which studies were performed over each sample.
Please, if possible, add a table with the sample description and the nomenclature used throughout the paper.
Author Response
Response to Reviewer 1 Comments
Point 1: It is stated that SEY is shown in Table 5 and Figure 9, but the data presented in both are different, especially in CMT03 and CTM30 samples.
Response 1: We are very sorry for this error caused by the inconsistency of axes and scales in the drawing process. We have replotted the SEY data and corrected the error in marking the sample number. We have modified Figure 9 and Table 5, incorporating the comments of another reviewer. The corresponding changes are in line 229-231 of the manuscript and are distinguished from the rest in the font format of Bold Green Italics.
Point 2: The Figure 4 label refers to (a) and (b) labels, which are not in the figure.
Response 2: We are very sorry for this error. We have distinguished (a) and (b) in Figure 4 and marked them on the graph. The revised Figure 4 is in line 158-160 of the manuscript.
Point 3: Please clarify how many samples were studied and state which studies were performed over each sample.
Response 3: All groups of samples listed in Table 2 were tested in this study. The size of the samples used for the test of surface flashover voltage in vacuum, secondary electron emission characteristics and resistivity were showd respectively in the “Sample preparation” section of the manuscript (Line 105-109). Bulk density, SEM, XRD, XPS tests do not have srict requirements for sample size, and the remaining samples from the above tests can be used.
Point 4: Please, if possible, add a table with the sample description and the nomenclature used throughout the paper.
Response 4: The sample number and main additives were shown in Table 2. And according to your suggestion, we have added the nomenclature of the samples used in this study. The corresponding changes are in line 108-109 of the manuscript and are distinguished from the rest in the font format of Bold Green Italics.

Reviewer 2 Report
Some problems should be issued.
1. Page 1, abstract, “ In the process of heat treatment, TiO2 in ceramics was reduced in a certain degree”. Does it mean Ti content is reduced and why?
2. Page 5, “Bulk density of Al2O3 ceramics with 1 wt. % TiO2 is highest, reaching 3.810 g/cm3. After that, the bulk density of Al2O3 ceramics decreases with the increase of the content of TiO2. Why?
3. The said that “Ti4+ replaces Al3+, which produces lattice deformation and a large number of cation vacancies”. Is there any evidence? For example, XRD peak shift.
4. The highest density of 3.810 g/cm3 is exhibited by CMT10, which is increased by 0.7% compared with CMT05. The increase is very slight. Except CMT30, the effects of density on the performance may be ignored.
5. Figure 9, what is CMT03?
6. What is the value of the SEY of the Cr/Mn doped Al2O3 ceramics CMT11? It is better to give the value in Table 5.
7. What are the influences of secondary electron emission of the ceramics on surface flashover voltage directly? Please clarify in introduction in detail.

Author Response
Point 1: Page 1, abstract, “ In the process of heat treatment, TiO2 in ceramics was reduced in a certain degree”. Does it mean Ti content is reduced and why?
Response 1: The sentence “In the process of heat treatment, TiO2 in ceramics was reduced in a certain degree” aims to express that TiO2 in ceramics undergoes a reduction reaction during the process of heat treatment at 1450 °C under wet hydrogen conditions, which is explained in the “Microstructure and phase composition” section of the manuscript.
Point 2: Page 5, “Bulk density of Al2O3 ceramics with 1 wt. % TiO2 is highest, reaching 3.810 g/cm3. After that, the bulk density of Al2O3 ceramics decreases with the increase of the content of TiO2. Why?
Response 2: The bulk density of the Cr/Mn/Ti doped Al2O3 ceramics was measured by the Archimedes' drainage method. The effect of TiO2 on the bulk density of Al2O3 ceramics is mainly due to: TiO2 can form the finite solid solution with Al2O3 when it is added to ceramics, which promotes the sintering of ceramics. But excessive TiO2 added to ceramics makes the fast or local abnormal grain growth during the ceramic sintering process, which is likely to cause the inclusion of pores in the grain and in the grain boundary can’t be eliminated in time, resulting in the reduction of the bulk density of ceramics. This part of the content is elaborated in the “Bulk density and micro morphology” section of the manuscript.
Point 3: The said that “Ti4+ replaces Al3+, which produces lattice deformation and a large number of cation vacancies”. Is there any evidence? For example, XRD peak shift.
Response 3: Thank you for your valuable advice. Accordingly, we anlayzed the local magnification of XRD patterns. We found that the characteristic peak of the Al2O3 at 2 Theta=35.15 shifts towards a small angle direction. This is due to the solid solution of TiO2 and Al2O3. Ti4+ replaces Al3+, which changes the cell parameters of Al2O3. We have modified the Figure 7. And the corresponding changes are in line 210-213 of the manuscript and are distinguished from the rest in the font format of Bold Green Italics.
Point 4: The highest density of 3.810 g/cm3 is exhibited by CMT10, which is increased by 0.7% compared with CMT05. The increase is very slight. Except CMT30, the effects of density on the performance may be ignored.
Response 4: Indeed, the bulk density of the CMT05 and CMT10 changed very little. However, afterwards, the bulk density of Al2O3 ceramics decreases with the increase of the content of TiO2. Although the percentage of the reduction is not significant, the 3.810 to 3.744 and then to 3.716 are still a relatively clear decreasing trend for the density of ceramic materials. It shows that the addition of TiO2 has an effect on the bulk density of the Al2O3 ceramics.
Point 5: Figure 9, what is CMT03?
Response 5: We are very sorry for the error in marking the sample number. We have modified Figure 9, incorporating the review comment of Point 6.
Point 6: What is the value of the SEY of the Cr/Mn doped Al2O3 ceramics CMT11? It is better to give the value in Table 5.
Response 6: Thank you for your valuable advice. We have added the SEY of the Cr/Mn doped Al2O3 ceramics CMT11 in Figure 9 and Table 5. This will make the presentation of our experimental results more complete and systemtatic.
Point 7: What are the influences of secondary electron emission of the ceramics on surface flashover voltage directly? Please clarify in introduction in detail.
Response 7: Thank you for your valuable advice. We have added some explanations about the influences of secondary electron emission of the ceramics on surface flashover voltage in introduction . And the corresponding changes are in line 41-48 of the manuscript and are distinguished from the rest in the font format of Bold Green Italics.

Reviewer 3 Report
The manuscript "Study of TiO2 on the voltage holdoff capacity of Cr/Mn doped Al2O3 ceramic in vacuum" by the team of authors: Dandan Feng, Xiaojing Wang, Shike Zhao, Baipeng Song, Guanjun Zhang is devoted to the actual problem of controlling the surface and volume breakdown resistance of insulating ceramics, which used in high voltage vacuum applications. The authors in section 1 - introduction, fully disclosed the existing issues in the area under study.
The manuscript is written in clear language and has a good logical structure. The conclusions presented by the authors are proven in the work.
Let me point out some nuances that could improve the readability of the work.
1. In Figure 5 it is not clear where the pores are included in the ceramic grain. Could you, as an explanation, show the main points in this figure with arrows?
2. The authors present the XPS spectrum of the Ti2p band, which convincingly shows (together with other data) that a partial reduction of the substance occurs. I think at this point it would also be useful to show what happens to oxygen bonds. The O2p states are quite informative given that there is some visual correlation to the observed effects.
3. The authors revealed the essence of expression (1), but did not provide any approximations by this expression. On what range of values is the given expression valid? Are there real estimates for E, or lambda?
4. Is it possible to give an approximate size of the region of coherent scattering of X-ray photons for the registered new phases?
5. The authors have said a lot about the mechanism of formation of secondary electron emission and its influence on the properties of surface breakdown, but no estimates of the work function for the studied ceramics have been given.
These nuances do not negate the significance of the work and the potential for further studies of the structural characteristics of such ceramics. I would like to draw the attention of the authors to the fact that such methods as cathodoluminescence in combination with EDX will allow a deeper understanding of the features of the microdefect structure, and stimulation of secondary electrons by temperature or UV light quanta will make it possible to understand which predominant states in the electronic structure form the valence band and how titanium is added shifts these energy states.
Author Response
Response to Reviewer 3 Comments
Point 1: In Figure 5 it is not clear where the pores are included in the ceramic grain. Could you, as an explanation, show the main points in this figure with arrows?
Response 1: Thank you very much for your suggestion. We have circled the main pores in Figure 5. Besides we explained them in line 188 of the manuscript and distinguished from the rest in the font format of Bold Green Italics.
Point 2: The authors present the XPS spectrum of the Ti2p band, which convincingly shows (together with other data) that a partial reduction of the substance occurs. I think at this point it would also be useful to show what happens to oxygen bonds. The O2p states are quite informative given that there is some visual correlation to the observed effects.
Response 2: Thank you very much for your suggestion. In this study, aluminum ion, chromium ion, manganese ion, titanium ion, silicon ion and calcium ion in the Cr/Mn/Ti Al2O3 doped ceramic, may all have bonds with oxygen ion, so the oxygen bonds in the ceramcs are very complicated. We tried to analyzed the oxygen bonds, and did not get any valuable information related to the change of valence state of titanium ion. We are very sorry, but in this regard, we will continue to conduct research in the future.
Point 3: The authors revealed the essence of expression (1), but did not provide any approximations by this expression. On what range of values is the given expression valid? Are there real estimates for E, or lambda?
Response 3: The expression (1) reveals the collisional ionization coefficient of the material, which refers to how easily an electron is collided and ionized. Where E is the strength of electric field applied to the material and is a variable. Lambda is the mean free path of electron. It is found that the mean free path of electron in Al2O3 is 50 ~ 100 nm. But the ceramic in this study is multiphase material, including Al2O3 crystals and Al-Si-Ca-O amorphous phase. There are also pores, impurities, defects, etc. in ceramics, which can scatter electrons, resulting in a limited electron free path. The more defects, the smaller the mean free path of electron. At present, the value range of mean free path of multiphase ceramic materials has not been found. In this study, the expression (1) is used to qualitatively analyze the variation trend of SEY of the Al2O3 ceramics.
Point 4: Is it possible to give an approximate size of the region of coherent scattering of X-ray photons for the registered new phases?
Response 4: The scanning range of the diffractometer we used during the XRD text is about 20 mm × 20 mm. The new phase discovered in the Al2O3 ceramic in this study was generated during the ceramic sintering process. Based on the scanning range of XRD testing and the generation process of new phases in ceramics, it can be considered that the registered new phases in the Al2O3 ceramic are uniformly distributed throughout the entire ceramic body.
Point 5: The authors have said a lot about the mechanism of formation of secondary electron emission and its influence on the properties of surface breakdown, but no estimates of the work function for the studied ceramics have been given.
Response 5: Thank you for your suggestion. We understand that the concept of work function is mostly applicable to metal material and alloy material. For insulator material, due to its non conductivity, it is not possible to test its work function using methods such as UPS. In this study, three gun method was used to measure the secondary electron emission coefficient of ceramics. The measuring gun emitted the primary electron to bombard the ceramic surface, and the secondary electrons were collected by a collector. The test conditions of each group of the samples are the same, so as to compare the SEY (secondary electron yield) of the Al2O3 ceramic. The SEY of ceramic directly affects the probability of SEEA (secondary electron avalanche) in vacuum.
Thank you again for the reviewer’s valuable feedback. As the reviewer said: “Methods as cathodoluminescence in combination with EDX will allow a deeper understanding of the features of the microdefect structure, and stimulation of secondary electrons by temperature or UV light quanta will make it possible to understand which predominant states in the electronic structure form the valence band and how titanium is added shifts these energy states.” In our subsequent research on high voltage resistant Al2O3 ceramics for vacuum device, defect characterization is a major and necessary subject to be studied. We will carefully consider the testing methods provide by the reviewer. And using these testing methods could charaterize the features of the microdefect structure, defects, etc. of the ceramic materials, providing strong theoretical support for the in-depth research of high voltage resistant ceramic materials in the future.
